# Evolution of an Iron-Detoxifying Protein: Eukaryotic and *Rickettsia* Frataxins Contain a Conserved Site Which Is Not Present in Their Bacterial Homologues

**DOI:** 10.3390/ijms232113151

**Published:** 2022-10-29

**Authors:** Rui Alves, Maria Pazos-Gil, Marta Medina-Carbonero, Arabela Sanz-Alcázar, Fabien Delaspre, Jordi Tamarit

**Affiliations:** Departament de Ciències Mèdiques Bàsiques, Facultat de Medicina, IRBLleida, Universitat de Lleida, 25001 Lleida, Spain

**Keywords:** iron, Friedreich’s ataxia, mitochondria, protein evolution

## Abstract

Friedreich’s ataxia is a neurodegenerative disease caused by mutations in the frataxin gene. Frataxin homologues, including bacterial CyaY proteins, can be found in most species and play a fundamental role in mitochondrial iron homeostasis, either promoting iron assembly into metaloproteins or contributing to iron detoxification. While several lines of evidence suggest that eukaryotic frataxins are more effective than bacterial ones in iron detoxification, the residues involved in this gain of function are unknown. In this work, we analyze conservation of amino acid sequence and protein structure among frataxins and CyaY proteins to identify four highly conserved residue clusters and group them into potential functional clusters. Clusters 1, 2, and 4 are present in eukaryotic frataxins and bacterial CyaY proteins. Cluster 3, containing two serines, a tyrosine, and a glutamate, is only present in eukaryotic frataxins and on CyaY proteins from the *Rickettsia* genus. Residues from cluster 3 are blocking a small cavity of about 40 Å present in *E. coli*’s CyaY. The function of this cluster is unknown, but we hypothesize that its tyrosine may contribute to prevent formation of reactive oxygen species during iron detoxification. This cluster provides an example of gain of function during evolution in a protein involved in iron homeostasis, as our results suggests that Cluster 3 was present in the endosymbiont ancestor of mitochondria and was conserved in eukaryotic frataxins.

## 1. Introduction

Friedreich’s ataxia is a rare, genetic neurodegenerative disease caused by mutations in the frataxin gene (FXN). In most patients, an expansion of GAA triplets is found in the first intron of this gene which cause a marked decrease in protein frataxin levels (below 30% of normal levels) [1]. Around 4% of patients are compound heterozygous for a GAA expansion and an FXN point mutation or deletion [2]. The major product of FXN is a polypeptide of 210 amino acids that is imported into mitochondria and processed by two sequential cleavages, which first produce an intermediate form (Fxn42-210) and subsequently the mature form (Fxn81-210). The intermediate form can be detected in certain tissues at low levels, and some authors have suggested that it could play specific roles. Alternative FXN transcripts not containing the mitochondrial targeting sequence have also been described [3]. Expression of these transcripts could be tissue-specific and their products may not be localized in the mitochondria. In this regard, one of these transcripts (FXN II) is mainly expressed in the cerebellum, and when transduced in human cell lines, the protein product shows a clear cytosolic localization [4]. A similar frataxin proteoform (isoform E) was identified as the major form in erythrocytes [5]. In mice, an extramitochondrial, truncated, proteoform of frataxin has also been observed [6]. Clearly, more studies are needed to clarify the functionality and tissue distribution of frataxin proteoforms.

Mature frataxin is a highly conserved protein and homologues can be found in most species. Bacterial frataxins are commonly denominated CyaY proteins. Its structure is formed by two alpha helixes joined by a series of antiparallel beta sheets (Figure 1). In human frataxin, a C-terminal tail is found inserted between the two alpha helixes. This tail protects the hydrophobic nucleus of the protein and may increase its stability, as the *Saccharomyces cerevisiae* frataxin (which lacks this C-terminal tail) presents lower stability than mammalian ones [7]. Another characteristic feature of frataxin structure is the presence of a group of exposed acidic residues which are located in the N-terminal alpha helix forming an acidic ridge. This acidic zone is able to bind iron and other divalent metals in a weak and non-specific electrostatic way [8]. Thus, in addition, a potential high affinity iron-binding site has been described. This site would involve His86, a residue located in the disordered N-terminal tail of the protein [9].

Frataxin deficiency causes a dysregulation in iron homeostasis. Iron deposits or accumulation have been observed in frataxin deficient *S. cerevisiae* [10], flies [11] and mammals [12]. Nevertheless, the precise functions of frataxin are not fully understood. The most widely accepted hypothesis is that frataxin participates in the biosynthesis of iron–sulfur centers by regulating the activity of cysteine desulfurase. This enzyme obtains sulfur from cysteine, which is then used for the biosynthesis of iron–sulfur clusters and other cofactors requiring this compound [13,14] Nevertheless, frataxin is not essential for iron–sulfur biogenesis, as deficiency in these cofactors is not always observed in frataxin-deficient cells [15]. Further, iron–sulfur deficiency in frataxin deficient *S. cerevisiae* is caused by metabolic remodeling, and it can be prevented by deleting key metabolic regulators [16] or by nitric oxide [17]. Also, frataxin proteoforms are observed in mature erythrocytes, which lack mitochondria and where iron–sulfur biogenesis may have already taken place [5]. Taken together, these observations suggest that frataxin could have additional functions beyond its role in iron–sulfur biogenesis. One of these functions could be iron detoxificaction by ferroxidation, as this activity has been observed in eukaryotic frataxins [18]. Interestingly, this activity is not present in *E. coli* CyaY, and it has been suggested that it could have been gained by frataxin during the appearance of early eukaryotes to compensate for the limited abundance of other iron-detoxifying proteins in mitochondria [19]. In this regard, Isaya and collaborators expressed a mitochondria-targeted form of CyaY in frataxin-deficient *S. cerevisiae* and observed that the bacterial protein complemented, to a large extent, the loss of yeast frataxin, but was less efficient in detoxifying excess labile iron during aerobic growth [20].

Based on these observations, we have hypothesized that residues involved in ferroxidase activity in eukaryotic frataxins would not be present in bacterial CyaY proteins. Therefore, they could be identified by an analysis of sequence conservation between frataxins and CyaY proteins, as functionally relevant residues in proteins are often conserved among most members of a protein family. In the present work, we have performed an analysis of amino acid conservation among frataxins and CyaY proteins. Moreover, we have classified highly conserved residues in different clusters according to their 3D distribution, as functionally important residues tend to cluster together in space forming three-dimensional residue clusters [21]. We have discovered that frataxin presents four conservation clusters. One of these clusters is formed by two serines, a tyrosine, and a glutamate placed close together. This site is present in eukaryotic and Ricketsia frataxins, but it is not conserved in most bacterial CyaY proteins. The presence of this site, which had not been noticed before, could provide new clues about frataxin function and explain the differences observed in iron-detoxifying activity between eukaryotic frataxins and prokaryotic CyaY proteins.

## 2. Results

### 2.1. Analysis of Mature Frataxin by ConSurf

In order to obtain a non-human driven analysis of the evolutionary conservation of frataxin amino acid sequence, we subjected the human mature frataxin sequence to analysis by the ConSurf server [22]. This bioinformatics tool estimates the evolutionary conservation of amino acid positions in a protein based on the phylogenetic relations between homologous sequences. We first performed the ConSurf analysis using the 3S4M protein databank entry against the Swiss-Prot database, restricting the analysis to sequences with an e-value higher than 0.0001 and presenting between 95% and 35% identity to the query sequence. This analysis will be referred as ConSurfID35, hereafter.

Thirteen unique representative frataxin sequences from different phylogenetic origins fulfilled the criteria: three mammals (*Rattus Norvegicus, Bos taurus,* and *Mus musculus*), three invertebrates (*Drosophila melanogaster, Caenorhabditis elegans,* and *Dictyostelium discoideum)*, a plant (*Arabidopsis thaliana*), two fungi (*Saccharomyces cerevisisae* and *Schizosaccharomyces pombe*) and three bacteria from the genus *Rickettsia* (*Rickettsia africae*, *Rickettsia felis*, and *Rickettsia prowazekii*). From the 129 amino acids present in the 3S4M sequence, 21 presented the maximum conservation score (s = 9), 48 residues presented scores between 1 and 8, and 60 residues had unreliable conservation scores due to insufficient data in the multiple sequence alignment. Figure 1A shows a ribbon representation of frataxin structure with the amino acids colored according to conservation scores. It can be observed that most of the highly conserved amino acids (s = 9, deep violet) are located in the beta-sheet. A multiple sequence alignment presenting the sequences used by ConSurf to calculate the conservation scores is shown in Figure 2.

This alignment is also color-coded by conservation scores and shows that most conserved residues are found in the central part of the sequence, which corresponds to the beta sheet of the protein. These highly conserved residues could be grouped in four different conservation clusters according to their spatial disposition. The four conservation clusters are indicated in the 3D structure shown in Figure 1B in colors red, blue, green, and orange, respectively. The first cluster consisted of non-polar residues oriented to the hydrophobic core of the protein (I145, I154, W173, L182, L186). The second one consisted of amino acids in the beta-sheet oriented to the external part of the protein (V131, T133, V144, N146, Q148, W155, S157). The third cluster consisted of four polar amino acids (Y143, S158, S161, and E189) which are placed close together. Finally, cluster 4 consisted of a single highly conserved glutamic acid (E111) in the N-terminal alpha helix. This residue forms part of the acidic ridge found in this region, which is composed of several acidic residues and which may be involved in metal binding. Although not precisely conserved (and therefore not detected by ConSurf), the presence of an acidic ridge can be observed in all the 13 sequences analyzed when the multiple sequence alignment is inspected in detail (see for instance positions E100/101, E108, or D115 in Figure 2). In clusters 1 to 4, we excluded glycines and prolines, as these residues tend to be conserved in order to maintain proper folding, rather than because they are part of a functional site. It is common to observe highly conserved isolated glycines or prolines, that enable turns or break helices.

To better understand how these clusters are conserved in bacterial CyaYs, we performed a new analysis including a larger number of sequences. To do so, we decreased the minimal percentage identity to 20%, while keeping the E-value threshold of 0.0001. This analysis is referred as ConSurf-ID20 and it includes 47 unique sequences, containing the 13 sequences used in ConSurf-ID35, the frataxin from the Microsporidian parasite *Encephalitozoon cuniculi*, and 34 bacterial CyaY proteins (the complete list can be found in Appendix A). In this analysis, 13 residues presented the maximum score (s = 9), while 22 residues had unreliable conservation scores due to insufficient data in the multiple sequence alignment. Figure 3 shows ribbon representations of frataxin structure with the amino acids colored according to conservation scores and conservation clusters.

A multiple sequence alignment presenting the sequences used by ConSurf to calculate the conservation scores is shown in Appendix A. Remarkably, the 13 highly conserved residues corresponded only to clusters 1 and 2. In cluster 1, three of the residues identified as highly conserved in the first analysis were also found highly conserved in this second analysis. These were I154, W173, and L186. Regarding the amino acids from cluster 2, most of them (with the exception of T133) were also identified as highly conserved in this second analysis. Residues from cluster 3 presented conservation scores below 9. Regarding cluster 4, E111 was substituted by D in most CyaY, thus keeping the acidic character. An acidic ridge can also be observed in all the sequences analyzed when the multiple sequence alignment is inspected in detail (Appendix A). We also observed three new residues (Q153, R165, Y166) which presented an s = 9 in this second analysis (but not in the first one). Q153 and R165 present the same orientation as amino acids from cluster 2, and could also be included in this group. The position of Y166 is more ambiguous. It is partially oriented towards the hydrophobic core of the protein and it could therefore be included in conservation cluster 1. The multiple sequence alignment shows that this position is always occupied by an aromatic residue, as a Tyr is found in most eukaryotic frataxins, while a Phe is found in most bacterial ones. Thus, we cannot exclude that the aromatic character of this residue may play a relevant role in frataxin function or structure.

### 2.2. Coevolution of Amino Acids from Cluster 3

The ConSurf-ID20 analysis show that cluster 3 is not conserved in bacterial CyaY proteins, prompting us to investigate in more detail the coevolution of the four amino acids forming the cluster (Table 1). Interestingly, these four amino acids are present in all eukaryotic frataxins, while absent in most CyaY proteins from bacterial origin, some of which contain one or two of these amino acids, but never Y143 and E189 simultaneously. The exceptions are the CyaY proteins from the *Rickettsia* genus, which present all the four amino acids from the cluster. Interestingly, phylogenomic analyses suggest that mitochondria most likely originated from the *Rickettsia*’s lineage [23]. Also, the presence of cluster 3 in *Rickettsia* is consistent with the higher degree of identity (and lower e-value) obtained in the alignment between human frataxins and CyaY from these bacteria (e-values and identity values form the alignments between the different sequences are available in the Appendix A). Thus, the four amino acids of cluster 3 present a marked coevolution which suggests that *Rickettsia* (and thus eukaryotic) frataxins may have experienced a gain of function during evolution.

### 2.3. Cluster 3 in the Structures of S. cerevisiae Frataxin, E. coli CyaY and Rickettsia CyaY

We used Matchmaker from Chimera to compare in more detail the structures of human mature frataxin with those of *S. cerevisiae* frataxin (2FQL) and *E. coli* CyaY (2EFF) in order to understand if the presence or absence of cluster 3 might have functional implications. These comparisons are shown in Figure 4.

*S. cerevisiae* frataxin and *E.coli* CyaY present a 3D structure highly similar to human frataxin, with the presence of the two alpha helixes and the central beta sheet. However, when we analyzed in detail the amino acids corresponding to cluster 3, we found marked differences between the eukaryotic frataxins and the bacterial CyaY. In both human and *S. cerevisiae* frataxin, the 3D disposition of the amino acids in cluster 3 was highly conserved, while those amino acids were not observed in CyaY. In this case, Tyr143 was replaced by an Ile, while two amino acids were observed close to the position of Glu189, a Gln and an Ala. This analysis confirms that cluster 3 is highly conserved in eukaryotes but it is absent in most bacterial CyaY proteins. If cluster 3 is structurally meaningful and was acquired by *Rickettsia*, then this cluster should also be structurally conserved in *Rickettsia*’s CyaY. We used Swismodel to investigate if this is so, by creating and analyzing structural models for CyaY from *R. tiphi*, *R. akari*, *R. bellii*, and *R. prowazekii*, using as a template the structure of frataxin from *Chaetomium thermophilum* (6FCO). All frataxins have predicted structures that are very similar to that of the human protein. In addition, these models show that cluster 3 is also spatially conserved with respect to the human structure (Figure 4C).

### 2.4. Structural and Functional Effects of Cluster 3

To investigate whether the presence or absence of cluster 3 might have a direct functional role in frataxin, we performed several in silico experiments.

First, we compare the structures of five frataxins (human, *S. cerevisiae*, *E. coli*, *R. akari*, and *R. prowazekii*) and identify potential grooves and cavities in the molecules. We found that *E. coli*’s frataxin has a small cavity of about 40 Å close to where cluster 3 residues are located, on the inside of the molecule. In frataxin from Rickettsia, this cavity moves closer to the outside of the molecule and is predicted to decrease its volume to 20 Å. In human and *S. cerevisiae* frataxin, the cavity disappears. The role of this cavity, if any, is unknown (Figure 5). To understand if the Y-S-S-E tetrad blocks the cavity observed in prokaryotic frataxins, we mutated the human frataxin structure, making it similar to that of *E. coli*: Y143→ I143, S158→T158, S161→Q161, and E189→A189. We optimized the mutated structure and reanalyzed it to find that a 38 Å cavity appeared, overlapping the one observed in *E. coli*’s frataxin (Figure 5C,D).

The existence of these cavities raises the possibility that either the cavity in prokaryotic frataxin or cluster 3 might be involved in binding. To investigate this issue, we used the COACH server [24], where three experimental and four modelled structures shown in Figure 4 were analyzed. This server predicts binding sites in proteins and assigns to each hit a C-score value between 0 and 1, where a higher score indicates a more reliable prediction. The server predicted a few metal binding sites with moderate confidence (C-scores between 0.2 and 0.5), but none of them involved amino acids from cluster 3 (no hits were observed for this cluster). The cavity observed in *E. coli* was predicted to bind cyclopentylacetic acid but with low confidence (C-score 0.01). We also investigated if a tetrad like that of cluster 3 can be found in active sites. We made an exhaustive analysis of the Mechanism and Catalytic site atlas [25], looking for active sites containing a Y-S-S-E tetrad. We found three enzymes (tryptophan synthase [unique id 383], beta lactamase class C [unique id 257], and NAD-ADP-ribosyl transferase [unique id 76]) with a Y-S-S-E tetrad involved in the catalytic activity. The catalytic roles of these residues are to serve as proton acceptor/donors, or hydrogen bond donor/acceptors. A more detailed structure comparison shows that the relative position of the residues in the tetrad is different between each of these three enzymes and human frataxin, and that additional residues are present in these active sites.

### 2.5. Similarities between Frataxins and Iron-Detoxifying Proteins

As it has been suggested that eukaryotic frataxins present higher iron-detoxification properties than bacterial CyaY proteins, we analyzed if cluster 3 presents some of the features of the ferroxidase centers from ferritin or from DPS, a bacterial protein involved in iron detoxification. These proteins present an iron-binding site composed of acidic residues and, in the case of DPS, histidines. Cluster 3 contains a single acidic residue, two serines, and a tyrosine (and no histidines). Serines and tyrosines are commonly found in metal sites coordinating Na^+^, Mg^2+^, K^+^, or Ca^2+^, but their presence in iron-binding sites is negligible [26]. Therefore, cluster 3 does not appear to be able to coordinate iron. Ferritins also present a highly conserved tyrosine within 5 Å of the ferroxidase center (Figure 6A). This residue provides a fourth electron to fully reduce molecular oxygen and prevent formation of reactive oxygen species [27]. In DPS, a highly conserved tryptophan residue is found at ∼3 Å from the iron-binding site and a nearby tyrosine residue is found in half of the bacterial sequences (∼8 Å from the tryptophan residue) (Figure 6B). These aromatic residues trap and dissipate potentially dangerous free electrons that may be generated during iron oxidation and prevent their diffusion into solution [28]. Interestingly, mammalian frataxins are rich in tryptophan and tyrosine residues. Human mature frataxin presents 7 tyrosines and 3 tryptophans, which represents a frequency of 7.7% (versus a ∼4.5% mean frequency of these amino acids in eukaryotic proteins [29]). Could these redox-active amino acids be required to dissipate free electrons generated during iron oxidation?

A visual inspection of the distribution of these residues in human frataxin indicates that most of them are separated by less than 10 Å, a distance that would allow electron transfer between them (Figure 6C). Tyrosine 143 (from cluster 3) is placed between Y123 and Y166 (∼10 Å away from each one). Several tyrosines and tryptophans are placed in the vicinity of Y123 or Y166, close enough to mediate electron transfer reactions between these residues and the putative iron-binding sites. In this regard, at least three putative iron-binding sites have been reported for frataxins: the acidic ridge (present in all frataxins and bacterial CyaYs), a high affinity site involving His86 (located in the disordered N-terminal tail), and a cavity between three frataxin subunits that was observed in the crystal structure of *S. cerevisiae* frataxin. This last iron-binding site would be formed by three aspartic residues (scD143) from three different frataxin polypeptide chains. This residue is conserved in human frataxin (D167), although its 3D disposition in the protein structure is slightly different. Interestingly, scD143 is ∼4 Å away from scW149, which has been identified by ConSurf analyses and included in cluster 1 (W173 in huFXN). ScY119 (Y143 in huFXN) is ∼17 Å from scW149, but an additional tyrosine placed in between (scY80) could facilitate electron transfer. Regarding human frataxin, several redox active aromatic residues are placed close to both D167 or the acidic ridge. It could be hypothesized that these residues would play a role in frataxin similar to the one played by the conserved tryptophan residue in DPS proteins, and that in eukaryotic frataxins Y143 would play an accessory role (as reported for the “nearby tyrosine” in DPS). Such tyrosine transfer chain could have been enriched with additional components in higher eukaryotes in order to provide enhanced antioxidant capacity to frataxin. Nevertheless, it is not clear which would be the role of E189, S158, S161 (the other residues of cluster 3) in this potential function.

### 2.6. Variants in the Human Population

Finally, we analyzed the conservation of amino acids from clusters 1 to 4 in the human population. Variants that do not cause disease, but which are present in the general population provide an estimation of the conservation of each amino acid position in the human genome. These natural variants are less common in those residues essential for the activity or stability of the protein. In contrast, disease-associated variants preferentially occur in functional sites [30]. Therefore, we performed two analyses: first, we searched gnomAD database for variants reported for mature frataxin in the general population; second, we analyzed the ConSurf scores from the 14 known pathological frataxin variants.

For the first analysis (Table 2), GnomAD v2.1.1 and v3.1.1 non-neuro datasets were used. These datasets contain data from samples that were not collected as part of a neurologic or psychiatric case/control study, or from samples collected as part of a neurologic or psychiatric case/control study but designated as controls [31]. Considering both datasets, natural variants were identified in 55 of the 129 residues (43%) composing mature frataxin. Variants observed in amino acids from clusters 1–4 are shown in Table 2, while a complete list of variants identified in mature frataxin is shown in Appendix A. We observed that 3 from the 14 known pathological variants (21%) were represented in these datasets. This proportion was similar to that observed for residues from conservation clusters 1–4, as only 4 from the 17 residues (24%) from these clusters presented variations in the datasets analyzed. Two of them were both serines from cluster 3, suggesting that the role of these residues in this potential functional site may be less essential than the role exerted by the Glu–Tyr dyad (which do not present variants). We also observed variations in two amino acids from the acidic ridge (E100 and D112), but not in the E111 residue identified as highly conserved in the ConSurfID35 analysis. Interestingly, both E100 and D112 present an adjacent acidic residue, suggesting that its substitution may have a mild effect on the acidic ridge (Appendix A). Regarding the second analysis (ConSurf scores from the pathological variants), Table 3 indicates the ConSurf scores of these residues in both ConSurf35 and ConSuf20 analyses, and also whether they belong or not to any of the conservation clusters defined in this work. From the 14 pathological point mutations described in the mature frataxin sequence, most of them (8/14, 57%) correspond to highly conserved amino acids belonging to clusters 1 and 2. Indeed, 7 of the 12 residues from these clusters presented pathological variants. No pathological point mutations were identified affecting clusters 3 and 4.

Overall, the information provided by these analyses suggest that the highly conserved amino acids identified in ConSurf analysis play fundamental roles in frataxin, as they are on average more conserved than expected in the human population, and highly represented among disease-associated variants.

## 3. Discussion

Conservation analysis is one of the most widely used methods for predicting functional sites in protein sequences, as these functionally relevant sites are usually conserved among members of a protein family. The power of this approach is improved when used in conjunction with structural information, as functionally important residues tend to cluster together in space. In the present work, we combined both sequence and structural conservation information to analyze the presence of potential functional sites in eukaryotic frataxins not present in bacterial CyaY proteins. We discovered that frataxin has four conservation clusters. Three of them (clusters 1, 2 and 4) are found in eukaryotic frataxins and bacterial CyaY proteins, suggesting their ancient evolutionary origin. A fourth cluster (cluster 3) is only observed in eukaryotic frataxins and on CyaY proteins from the *Rickettsia* genus.

Cluster 1 is formed by non-polar residues oriented to the hydrophobic core of the protein, which may be involved in maintaining the 3D structure of the protein. A hypothetical antioxidant function may also be envisaged for W173 and W168, similar to the one exerted by tryptophan in DPS. Four-point mutations affecting residues from this cluster have been identified in FA patients. The most studied one is the I154F substitution, which is believed to decrease protein stability and results in decreased content of mature frataxin [32]. The W173G mutation has been reported to inhibit the processing of frataxin into its mature form by an unknown mechanism [33]. The functional consequences of the two other mutations (affecting L182 and L186) have not been investigated in detail. Cluster 2 is formed by amino acids in the beta-sheet, oriented to the external part of the protein. In this second cluster, four pathological mutations have been identified, the W155R substitution being the most extensively investigated. This mutation has been shown to decrease the interaction of frataxin with ISD11 [34], suggesting that cluster 2 may be involved in protein–protein interactions required for frataxin function.

Cluster 3 is only observed in eukaryotic frataxins and on CyaY proteins from the *Rickettsia* genus and had not been noticed before. The conservation of this cluster in *Rickettsia* is also an interesting finding, as it has been hypothesized that the mitochondrial endosymbiont which was at the origin of mitochondria was an obligate aerobe similar to modern *Rickettsia* species [23]. This suggests that cluster 3 was present in the mitochondrial ancestor and is conserved in eukaryotic frataxins. To uncover the function of this cluster, we have used some in silico approaches to provide some hypothesis about its function. We have used COACH and the Mechanism and Catalytic site atlas to predict if residues from cluster 3 could be involved in specific binding or catalytic activity. While some hits were found, the confidence in the prediction is moderate to low. We have also found that this cluster is blocking a small cavity of about 40 Å present in *E. coli*’s CyaY. None of these analyses has provided a working hypothesis about the function of cluster 3, and we cannot exclude that these group of amino acids may have a structural function. In this regard, tyrosines are known to contribute to protein structure stability [35]. Nevertheless, we have observed that human frataxin is rich in tryptophan and tyrosine residues, two residues which are known to readily undergo radicalization. This characteristic allows them to mediate electron transfer reactions in proteins and act as internal antioxidants, as it has been demonstrated in ferritin and bacterial DPS, two proteins involved in iron detoxification. In ferritin, a conserved tyrosine provides a fourth electron to fully reduce molecular oxygen preventing the formation of reactive oxygen species as a by-product of the ferroxidation reaction [27]. In DPS, tryptophan and tyrosine residues near the iron-binding site contribute to limit release of hydroxyl radicals in solution after iron oxidation [28]. An enriched presence of tyrosine and tryptophan residues has also been observed in transmembrane domains of membrane proteins, and has been linked to the antioxidant properties of these amino acids [36]. Therefore, it can be speculated that tyrosine from cluster 3 and the other redox-active aromatic residues present in frataxin could be performing an antioxidant function that would provide eukaryotic frataxins with an increased ferroxidase activity. Wet lab experiments will be required to test this hypothesis.

No pathological mutations in cluster 3 have been identified. This could be due to random chance, but it could also indicate that mutations in amino acids from clusters 1 and 2 have a more marked effect on protein function or stability than those on clusters 3. For instance, as indicated before, I154F and W173G mutations result in decreased content of mature frataxin, which may cause a loss in frataxin activity below the pathological threshold required to trigger FA. In contrast, individuals presenting a mutation in cluster 3 in one allele and a GAA expansion in another allele (which usually produces a certain amount of functional frataxin) may present frataxin activity above pathological threshold. In this regard, it is worth reminding that studies performed in *Δyfh1 S. cerevisiae* expressing CyaY protein indicate that this protein may partially complement the loss of frataxin [20]. Nevertheless, the absence of variants for Y143 and E189 in the gnomAD database is consistent with the hypothesis that these residues perform a relevant function.

Cluster 4 consists of a single glutamic acid (E111). This residue is the most conserved one from the frataxin acidic ridge. It is also conserved among human populations, as variants have not been reported in the gnomAD database. Both observations suggest that this residue could play a more relevant function than other residues from the acidic ridge. Nevertheless, studies in *S. cerevisiae* have shown that mutation of this residue (E89 in *S. cerevisiae*) does not cause alterations in iron homeostasis. Actually, several residues from the acidic ridge may be simultaneously mutated in order to seriously compromise frataxin function [37]. This observation could explain the absence of pathological mutants in the acidic ridge. The precise positions of the other acidic amino acids from the acidic ridge are less conserved, and therefore they are not detected by ConSurf. We have also observed that the two residues from the acidic ridge presenting variation in human GenomAD database (E100 and D112) present a contiguous acidic residue, suggesting that they could be playing an accessory role in frataxin function (Appendix A).

Biochemical in vitro studies have also indicated that human frataxin and *E. coli* CyaY may play a different role in the iron–sulfur biogenesis process. Frataxin activates cysteine desulfurase and iron–sulfur biogenesis, while the *E. coli* CyaY inhibits this same process. Enzyme kinetic experiments revealed that activation or inhibition by the frataxin homologue was determined by which cysteine desulfurase (eukaryotic or procariotic) was present in the reaction mixture and not by the identity of the frataxin homologue [38]. Therefore, the presence of cluster 3 in eukaryotic frataxins does not appear to be the reason for this functional difference between bacterial CyaY proteins and eukaryotic frataxins.

## 4. Materials and Methods

### 4.1. Conservation Analysis

Conservation analysis was performed using The Consurf Server available at https://consurf.tau.ac.il/ (accessed from 23 September 2019 to 8 October 2020) [22]. We obtained the frataxin sequence from the PDB structure 3S4M, as ConSurf server uses PDB entries as input information, a characteristic that allows the representation of the conservation scores in the 3D structure of the protein. Among the different human frataxin structures available in the protein data bank, the 3S4M entry presents the sequence most representative of mature frataxin (Fxn81-210). This sequence starts at residue 82 (G82) and from this residue onward is identical to Uniprot Q16595-1 entry (which corresponds to human frataxin isoform 1). Settings used for the search were “Analyze Amino-Acids”; protein structure: PDB code 3S4M; Chain Identifier: A; no Multiple Sequence Alignment (MSA) uploaded; Protein database: SwissProt; Homolog search algorithm: HMMER; HMMER E-value: 0.0001; No. of HMMER Iterations: 1; Maximal %ID Between Sequences: 95; Minimal %ID For Homologs: 20 (for ConSurf20ID) or 350 (for ConSurf35ID); Number of sequences: a maximum of 150 sequences that sample the list of homologues to the query (sample 150 sequences in equal intervals from the list of protein sequences fulfilling the indicated criteria).

### 4.2. Protein Molecular Graphics and Analyses

Molecular graphics and analyses were performed with UCSF Chimera [39], developed by the Resource for Biocomputing, Visualization, and Informatics at the University of California, San Francisco. Structural models for CyaYs from *Rickettsias* were built using SwissModel [40] and using, as a template, the structure of frataxin from Chaetomium thermophilum (6FCO).

## 5. Conclusions

In summary, by an analysis of amino acid sequence conservation, and taking into account the spatial distribution of highly conserved residues, we have discovered the presence of four clusters of highly conserved amino acids in frataxin. Cluster 3 is present in eukaryotic frataxins and *Rickettsia* CyaY proteins, but not in most bacterial CyaYs. This potential functional site could provide new clues about frataxin function and explain the differences in ferro-oxidase activity observed between eukaryotic frataxins and prokaryotic CyaY proteins.

## Figures and Tables

**Figure 1 ijms-23-13151-f001:**
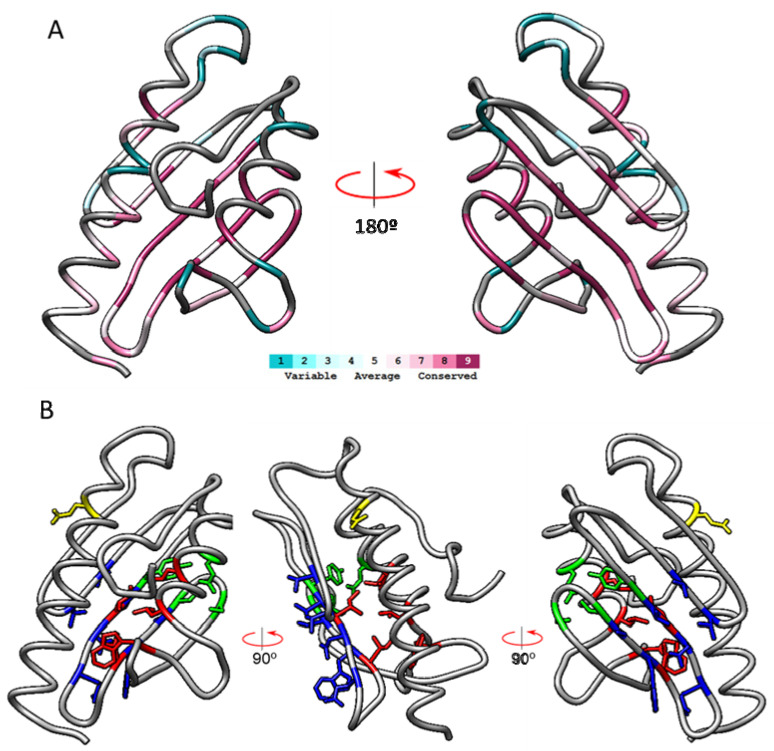
Frataxin structure (PDB ID:3S4M) indicating the conserved amino acids identified in ConSurfID35 analysis. In (**A**), frataxin structure is shown in ribbons and colored according to the conservation scores calculated by ConSurf. In (**B**), highly conserved amino acids (ConSurf score = 9) are shown in sticks and colored according to the four conservation clusters defined: cluster 1 red, cluster 2 blue, cluster 3 green, and cluster 4 yellow.

**Figure 2 ijms-23-13151-f002:**
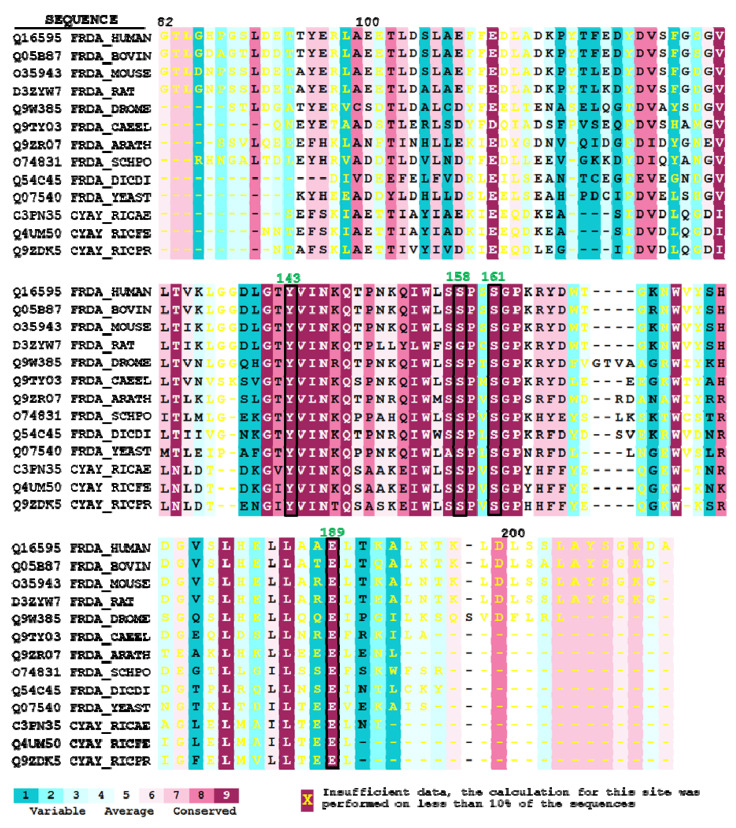
Multiple sequence alignment of Frataxin sequences used in ConSurfID35 analysis. Residues are colored according to the conservation scores calculated by ConSurf. The input pdb sequence corresponds to the sequence of human mature frataxin extracted from pdb entry 3S4M. Residues from conservation cluster 3 are boxed, and its position in the human frataxin sequence indicated in green numbers. Uniprot codes and entry names are indicated for each sequence.

**Figure 3 ijms-23-13151-f003:**
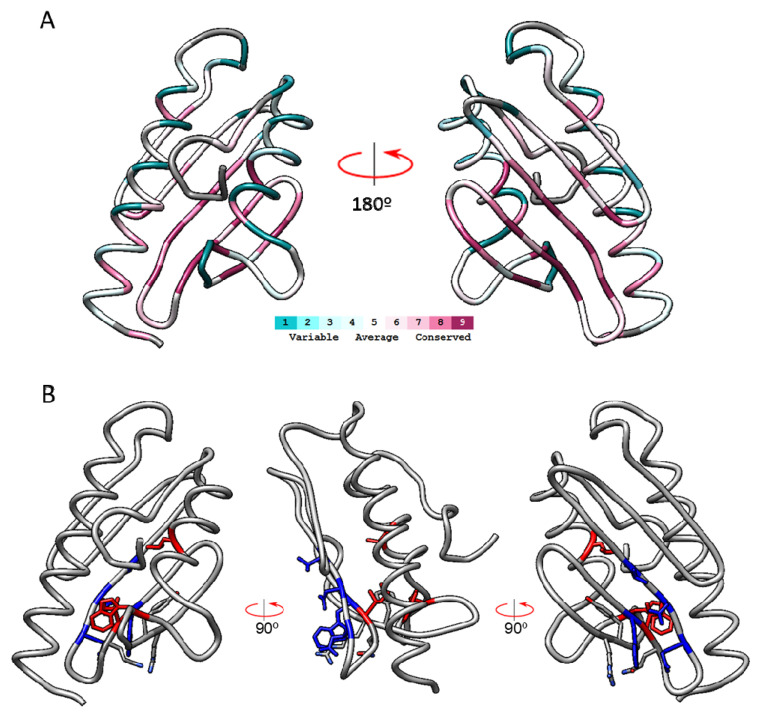
Frataxin structure (PDB ID:3S4M) indicating the conserved amino acids identified in ConSurf-ID20 analysis. In (**A**), frataxin structure is shown in ribbons and colored according to the conservation scores calculated by ConSurf. In (**B**), highly conserved amino acids (ConSurf score = 9) are shown in sticks and colored according to the conservation clusters defined: cluster 1 red, cluster 2 blue. Amino acids from cluster 3 and cluster 4 are not shown, as they are not highly conserved in this second analysis (which includes many bacterial CyaY proteins). Q153, R165, and Y166, which presented an s = 9 in this second analysis but not in the first one, are colored by element (CPK coloring).

**Figure 4 ijms-23-13151-f004:**
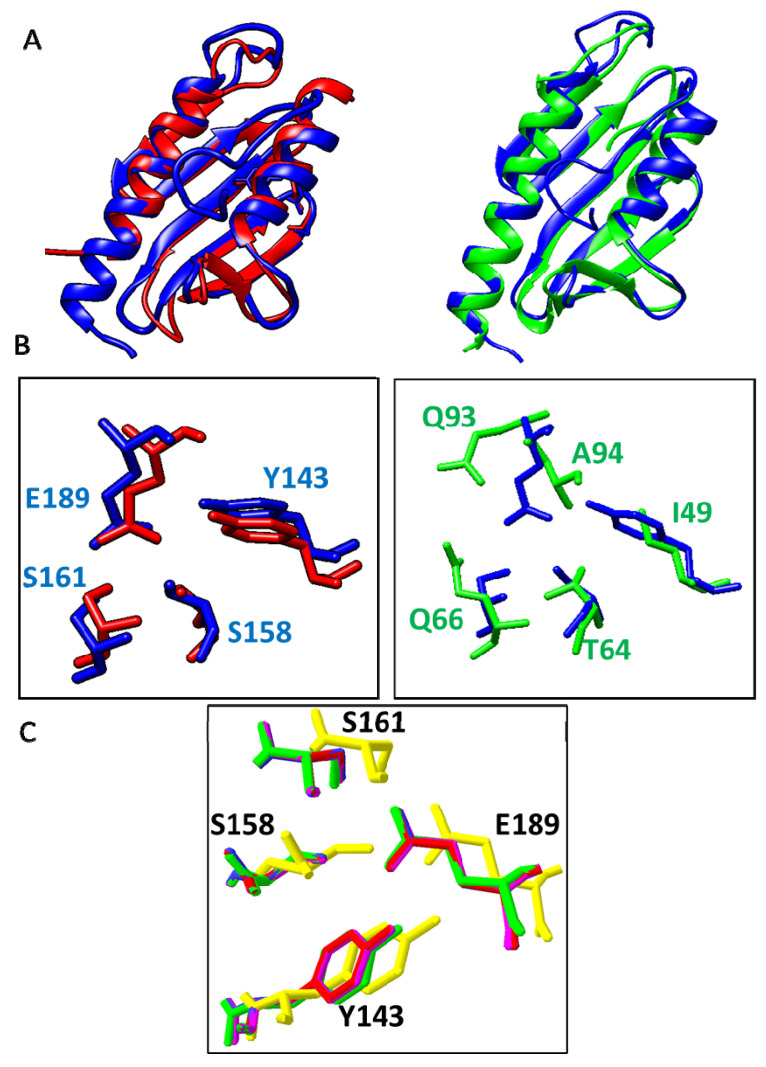
Comparison of the 3D structures of human frataxin (blue, PDB ID:3S4M) with those of *S. cerevisiae* frataxin (red, PDB ID:3OER), and *E. coli* CyaY (green, PDB ID:2EFF). (**A**) Superposed representation of the indicated structures. (**B**) Amino acids forming cluster 3 in the superposed structures from human (blue) and *S. cerevisiae* (red) frataxins (left panel), and in those from human frataxin (blue) and *E. coli* CyaY (green), right panel. (**C**) Cluster 3 in structural models for CyaY from *R. tiphi* (blue), *R. akari* (mauve), *R. prowazekii* (red), and *R. bellii* (yellow). Residues from human frataxin are shown in green and its sequence position indicated.

**Figure 5 ijms-23-13151-f005:**
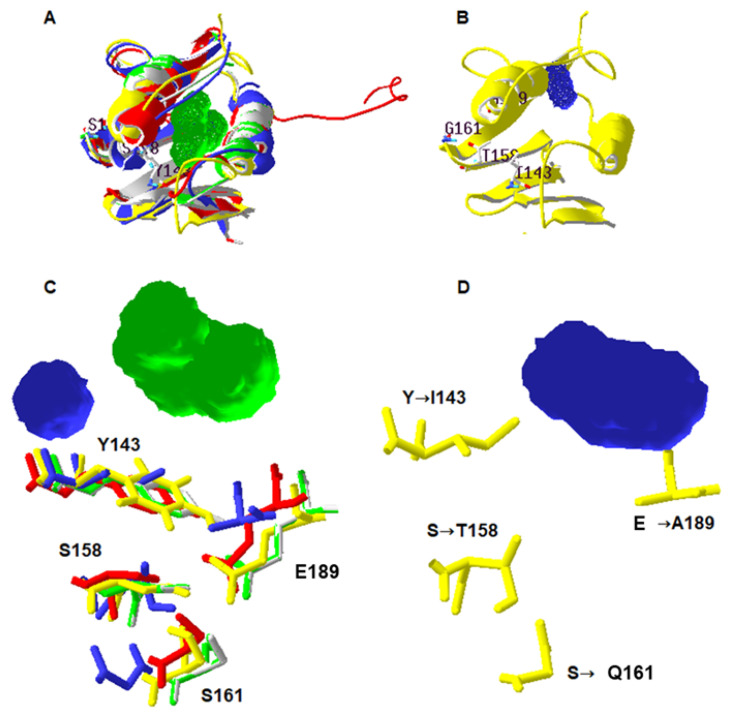
Structural analysis of frataxin. (**A**,**C**)—Superimposed structures of frataxin from human (yellow), *E. coli* (blue), *S. cerevisiae* (red), *R. prowazekii* (green), and *R. tiphi* (gray). (**B**,**D**)—Optimized structure of mutated human frataxin. The cavity in prokaryotic frataxins is represented by the green blob in panel (**C**). In the same panel, we also represent the cavity found in Rickettsia’s frataxin. The blue blob in panels (**B**,**D**) represents the cavity that arise in human frataxin when cluster three residues are mutated to those of *E. coli*. The blue cavity in panel (**D**) overlaps with the green cavity in panel (**C**) to a high extent.

**Figure 6 ijms-23-13151-f006:**
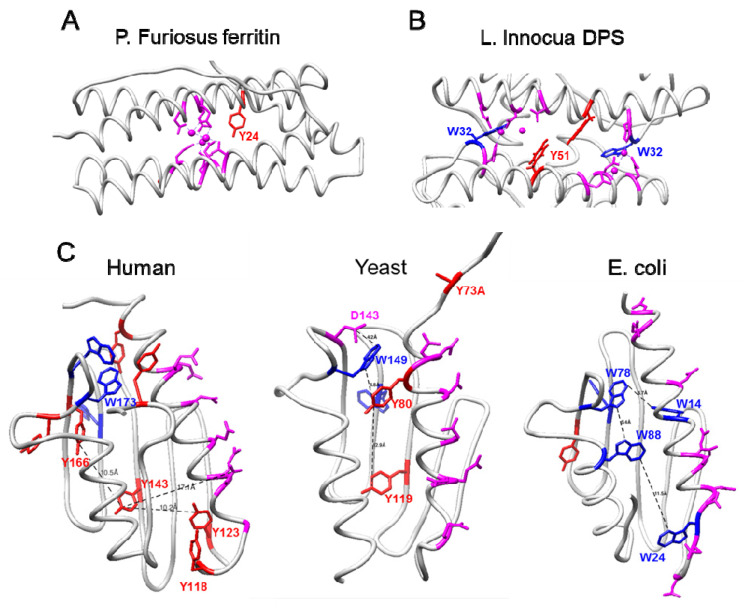
(**A**) Ribbon representation of one subunit of ferritin from *Pyrococcus furiosus* (PDB ID: 2JD7). A highly conserved tyrosine (Y24) is present in the vicinity (4–5 Å) of the ferroxidase center (shown in magenta) which is rich in acidic residues. (**B**) In *Listeria innocua* DPS (PDB ID: 6HX2), the ferroxidase center is formed at the interface of two subunits. A highly conserved tryptophan (W32) is present near the ferroxidase center (shown in magenta), while a nearby tyrosine (Y51) is placed 7.5 Å away from the tryptophan residue. (**C**) Ribbon representations of human and *S. cerevisiae* frataxins, and CyaY from *E. coli*. Tryptophans are shown in blue, tyrosines in red, while putative iron-binding acidic residues are shown in magenta. Those residues mentioned in the main text are indicated. In 3OER *S. cerevisiae* structure, Y73 is replaced by an Alanine.

**Table 1 ijms-23-13151-t001:** Presence of the four amino acids from conservation cluster 3 in the 47 sequences analyzed. Presence of the indicated residue in the sequence is indicated by “yes”. The one-letter code is used when other residues are found in that position.

SEQUENCE	Y143	S158	S161	E189	Organism
Input Sequence (HUMAN)	yes	yes	yes	yes	*Homo sapiens*
FRDA_BOVIN	yes	yes	yes	yes	*Bos taurus*
FRDA_MOUSE	yes	yes	yes	yes	*Mus musculus*
FRDA_RAT	yes	G	yes	yes	*Rattus norvegicus*
FRDA_DROME	yes	yes	yes	yes	*Drosophila melanogaster*
FRDA_CAEEL	yes	yes	yes	yes	*Caenorhabditis elegans*
FRDA_ARATH	yes	yes	yes	yes	*Arabidopsis thaliana*
FRDA_SCHPO	yes	yes	yes	yes	*Schizosaccharomyces pombe*
FRDA_DICDI	yes	yes	yes	yes	*Dictyostelium discoideum*
FRDA_YEAST	yes	yes	yes	yes	*Saccharomyces cerevisiae*
CYAY_RICAE	yes	yes	yes	yes	*Rickettsia africae*
CYAY_RICFE	yes	yes	yes	yes	*Rickettsia felis*
FRDA_ENCCU	yes	yes	T	yes	*Encephalitozoon cuniculi*
CYAY_NEIMB	I	A	G	A	*Neisseria meningitidis serogroup B*
CYAY_JANMA	I	A	G	M	*Janthinobacterium sp.*
CYAY_RICBR	yes	yes	yes	yes	*Rickettsia bellii*
CYAY_RICAH	yes	yes	yes	yes	*Rickettsia akari*
CYAY_AZOVD	L	A	G	V	*Azotobacter vinelandii*
CYAY_RICTY	yes	yes	yes	yes	*Rickettsia typhi*
CYAY_PSEPG	L	D	G	I	*Pseudomonas putida*
CYAY_TOLAT	I	T	N	A	*Tolumonas auensis*
CYAY_SALPK	I	T	yes	A	*Salmonella paratyphi A*
CYAY_PSEPF	L	A	G	I	*Pseudomonas fluorescens*
CYAY_THISH	L	yes	G	-	*Thioalkalivibrio sulfidiphilus*
CYAY_SALA4	I	T	G	A	*Salmonella agona*
CYAY_AERS4	V	T	N	A	*Aeromonas salmonicida*
CYAY_ECO81	I	T	G	A	*Escherichia coli O81*
CYAY_EDWI9	I	T	G	-	*Edwardsiella ictaluri*
CYAY_ALISL	I	yes	G	yes	*Aliivibrio salmonicida*
CYAY_CHLT3	yes	E	N	L	*Chloroherpeton thalassium*
CYAY_PSEF5	V	A	G	I	*Pseudomonas fluorescens*
CYAY_VIBCH	I	yes	G	yes	*Vibrio cholerae serotype O1*
CYAY_PSEU2	L	A	G	F	*Pseudomonas syringae pv. syringae*
CYAY_PSESM	L	A	G	M	*Pseudomonas syringae pv. tomato*
CYAY_VIBPA	I	yes	G	yes	*Vibrio parahaemolyticus serotype O3:K6*
CYAY_VIBVU	I	yes	G	yes	*Vibrio vulnificus*
CYAY_ALIFM	I	yes	G	yes	*Aliivibrio fischeri*
CYAY_PSEPW	L	D	G	I	*Pseudomonas putida*
CYAY_VIBTL	I	yes	G	yes	*Vibrio atlanticus*
CYAY_PSEFS	L	A	G	L	*Pseudomonas fluorescens*
CYAY_PHOLL	I	T	G	A	*Photorhabdus laumondii subsp. laumondii*
CYAY_HERAR	I	T	G	-	*Herminiimonas arsenicoxydans*
CYAY_CROS8	I	T	G	A	*Cronobacter sakazakii*
CYAY_SALNS	I	T	G	A	*Salmonella newport*
CYAY_PSEAE	L	A	G	-	*Pseudomonas aeruginosa*
CYAY_ENT38	I	T	G	A	*Enterobacter sp.*
CYAY_DICCH	I	T	G	A	*Dickeya chrysanthemi*

**Table 2 ijms-23-13151-t002:** Variants and its frequency observed in the indicated GnomAD non-neuro databases in amino acids from cluster 1 to 4 and from the acidic ridge. Frequency is indicated in ×10^5^. Variant observed is indicated in one-letter amino acid code.

Cluster	Position	Variant GnomAD	Freq v3.1.1	Freq v2.1.1	Pathological Variant
**C1**	I145	No			No
	I154	F	0.742	0.872	Yes
		V	0.742	9.41	No
	W173	No			Yes
	L182	No			Yes
	L186	No			Yes
**C2**	V131	No			No
	T133	A	0.742	1.922	No
	V144	No			No
	N146	No			Yes
	Q148	No			Yes
	W155	No			Yes
	S157	No			No
**C3**	Y143	No			No
	S158	P	0.742	4.703	No
	S161	R	0.742		No
	E189	No			No
**C4**	E111	No			No
**Acidic Ridge**	D112	Y		0.961	No
		H	0.742		No
	E100	A	9.65	24.505	No

**Table 3 ijms-23-13151-t003:** ConSurf scores of Frataxin pathological point mutations.

Residue	Mutation	rs ID	ConSurf Score (35ID)	ConSurf Score (20ID)	Conservation Cluster
LEU106	S	rs104894105	7	7	-
ASP122	Y	rs142157346	1	6	-
GLY130	V	rs104894107	6	8	-
ASN146	K	rs146818694	9	9	Cluster 2
GLN148	R	rs140472905	9	9	Cluster 2
ILE154	F	rs104894106	9	9	Cluster 1
TRP155	R	rs138471431	9	9	Cluster 2
LEU156	P	rs143340609	6	7	-
ARG165	C	rs138034837	7	9	Cluster 2
TRP173	G	rs56214919	9	9	Cluster 1
LEU182	F	rs139616452	9	8	Cluster 1
	H	rs149335881			
HIS183	R	rs144610605	3	1	-
LEU186	R	rs148443992	9	9	Cluster 1
LEU198	R	rs144104124	5	5	-

## Data Availability

Protein structures analyzed in this study can be accessed at https://www.rcsb.org/, accessed on 30 September 2022, while variants and mutations analyzed can be accessed at https://gnomad.broadinstitute.org/, accessed on 30 September 2022.

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
