# Peer review of "Evolution of an Iron-Detoxifying Protein: Eukaryotic and Rickettsia Frataxins Contain a Conserved Site Which Is Not Present in Their Bacterial Homologues"

_ijms, 2022, doi:10.3390/ijms232113151_

Round 1
Reviewer 1 Report
The manuscript by Alves et al. "Evolution of an iron-detoxifying protein: Eukaryotic and Rickettsia frataxins contain a conserved site which is not present in their bacterial homologs," addressed the conservation of structural properties among human frataxin and its bacterial homologs, CyaY proteins. Frataxin plays an essential role in iron mitochondrial metabolism, and frataxin deficiency due to mutations results in Frederic's ataxia (FRDA), a severe and lethal inherited disease. Eucaryotic frataxin and bacterial CyaY proteins are different in functions and properties. The authors performed sequence analysis of human frataxin and bacterial CyaY proteins to elucidate the structure/function relationship in frataxin and its bacterial homologs. The authors identified four conserved clusters in frataxins and CyaY. When clusters 1,2 and 4 are presented in all frataxins and CyaY proteins. However, cluster 3 was present only in frataxins and Rickettsia CyaY proteins. The authors used several in silico analysis tools to examine the structural properties of this cluster and its relation to mutations of FRDA. The manuscript offers a novel approach to frataxin and CyaY structural analysis and new clues about frataxin function. Here are a few minor comments.
Introduction.
The authors give an extensive overview of recent research on frataxin, and including newer references on extra-mitochondrial frataxin would benefit the introduction like as PMID: 33158039, PMID: 32978498.
I recommend improving the visual presentation of the data in figure 4c. The color-coded grey sequence for R.tiphi is barely visible.
Author Response
Comment: The authors give an extensive overview of recent research on frataxin, and including newer references on extra-mitochondrial frataxin would benefit the introduction like as PMID: 33158039, PMID: 32978498.
Answer: The references suggested by the reviewer have been included and commented in lines 37-45
Comment: I recommend improving the visual presentation of the data in figure 4c. The color-coded grey sequence for R.tiphi is barely visible.
Answer: Positions of the indicated residues in human, R. tiphi, R. akari, and R. prowazekii frataxins are very similar, and it is difficult to distinguish all of them when superposed. Nevertheless, we agree that grey color was not the most appropriate and colors have been changed to improve presentation.
Reviewer 2 Report
This is a well-written and interesting work supported by quality bioinformatics-based analyses. The Authors have demonstrated the potential importance of four clusters of residues in the important frataxin family proteins, one of which is newly described in this work. Especially significant are the findings related to cluster 3, which differentiates Rickettsial and eukaryotic frataxins from bacterial frataxin-like proteins. This work naturally leads to new questions as stated throughout the paper by the Authors (and as is the case in many bioinformatics-based papers) and will likely lead to future functional studies related to the role for these four clusters in the biochemistry of frataxin-like proteins.
Minor changes include:
(1) Consider italicizing all species names (e.g. E. coli to E. coli).
(2) In Figure 1, consider changing either red or red-orange to a different color; it is somewhat difficult to tell the difference between these two shades in Figure 1B.
(3) There are several places in the paper where 'amino acids' is rendered 'aminoacids', a quick search and correction should suffice.
(4) Line 160: 'where' should be 'were'.
(5) Lines 192-3: I believe this statement should be reversed; I believe mitochondria are thought to have originated from Rickettsia and not the other way around.
(6) Line 251: See comment 4 above.
Author Response
Comment 1: Consider italicizing all species names (e.g. E. coli to E. coli).
Answer: corrected, we have performed this action in all species mentioned
Comment 2: In Figure 1, consider changing either red or red-orange to a different color; it is somewhat difficult to tell the difference between these two shades in Figure 1B.
Answer: Figure legend has been corrected, as colors were not correctly indicated (cluster 4 is shown in yellow, not in orange). We apologize for the mistake
Comment 3: There are several places in the paper where 'amino acids' is rendered 'aminoacids', a quick search and correction should suffice.
Answer: OK, corrected
Comment 4: Line 160: 'where' should be 'were'.
Answer: corrected (now in line 165)
Comment 5: Lines 192-3: I believe this statement should be reversed; I believe mitochondria are thought to have originated from Rickettsia and not the other way around.
Answer: Sentence has been changed to: “phylogenomic analyses suggest that mitochondria most likely originated from the Rickettsias lineage” (line 198-99), which reflects better the conclusions of the work cited.
Comment 6: Line 251: See comment 4 above.
Answer: corrected (now in line 259)